# The First Large Identification of 3ANX and NX Producing Isolates of *Fusarium graminearum* in Manitoba, Western Canada

**DOI:** 10.3390/toxins17010045

**Published:** 2025-01-17

**Authors:** Maria Antonia Henriquez, Srinivas Sura, Sean Walkowiak, David Kaminski, Anne Kirk, Mark W. Sumarah, Parthasarathy Santhanam, Nina Kepeshchuk, Jules Carlson, E. RoTimi Ojo, Pam de Rocquigny, Holly Derksen

**Affiliations:** 1Morden Research and Development Centre, Agriculture and Agri-Food Canada, Morden, MB R6M 1Y5, Canada; parthasarathy.santhanam@agr.gc.ca (P.S.); nina.kepeshchuk@agr.gc.ca (N.K.); jules.carlson@agr.gc.ca (J.C.); 2Canadian Grain Commission, Grain Research Laboratory, Winnipeg, MB R3T 2N2, Canada; sean.walkowiak@grainscanada.gc.ca; 3Manitoba Agriculture, 65-3rd Avenue NE, Carman, MB R1N 1Y7, Canada; davkam5100@gmail.com (D.K.); anne.kirk@gov.mb.ca (A.K.); timi.ojo@gov.mb.ca (E.R.O.); pam@mbcropalliance.ca (P.d.R.); holly.derksen@upl-ltd.com (H.D.); 4London Research and Development Center, Agriculture and Agri-Food Canada, London, ON N5V 4T3, Canada; mark.sumarah@agr.gc.ca; 5Manitoba Crop Alliance, 38-4th Avenue NE, Carman, MB R1N 1Y7, Canada; 6UPL AgroSolutions Canada, 2-400 Michener Road, Guelph, ON N1K 1E4, Canada

**Keywords:** mycotoxins, trichothecenes, *Tri1*, NX, 3ANX, NX-2, NX-3, *Fusarium graminearum*, wheat, high-resolution mass spectrometry

## Abstract

Fusarium head blight, caused by *Fusarium graminearum*, continues to be one of the most important and devastating fungal diseases on cereal grains including wheat, barley, and oat crops. *F. graminearum* produces toxic secondary metabolites that include trichothecene type A and type B mycotoxins. There are many variants of these toxins that are produced, and in the early 2010s, a novel type A trichothecene mycotoxin known as 3ANX (7-α hydroxy,15-deacetylcalonectrin) and its deacetylated product NX (7-α hydroxy, 3,15-dideacetylcalonectrin) were identified in Minnesota, USA. In the current study, a total of 31,500 wheat spikes over a period of 6 years (2015–2020) within Manitoba, Canada, were screened for the *F. graminearum* pathogen, which accounted for 72.8% (2015), 98.3% (2016), 71.9% (2017), 74.4% (2018), 92.6% (2019), and 66.1% (2020) of isolations. A total of 303 *F. graminearum* isolates, confirmed through sequencing of the ribosomal intergenic spacer, were further investigated for variation in the gene *Tri1*, which was previously associated with the production of the NX toxin, as well as the accumulation of mycotoxins. A subset of these isolates, consisting of 73 isolates, which tested positive or negative for the NX-*Tri1*-F/R assay in this study, were cultured in vitro using rice media. Mycotoxins were quantified in these samples using mass spectrometry. Using the same rice culture, genomic DNA was isolated, and the *Tri1* coding sequence along with its flanking regions (upstream and downstream of the *Tri1* gene) was amplified and sequenced. Deoxynivalenol (DON) accumulated in 96% of the cultures from these isolates, while 3-acetyl deoxynivalenol (3ADON) and 3ANX mycotoxins accumulated in 66% and 63%, respectively. Nivalenol, 15-acetyl deoxynivalenol, and NX mycotoxins were detected in 62%, 36%, and 19% of samples, respectively. A significant correlation was observed between 3ADON and 3ANX (r^2^ = 0.87), as well as between DON and 3ANX (r^2^ = 0.89). This study highlights the first large identification of 3ANX- and NX-producing isolates of *F. graminearum* in Western Canada. In addition, it is the first identification of 15ADON chemotypes producing 3ANX in Western Canada and the first identification of 3ANX and NX-producing isolates in Manitoba, collected from wheat samples.

## 1. Introduction

Fusarium head blight (FHB), caused primarily by *Fusarium graminearum,* is one of the most prevalent and devastating fungal diseases on cereal grains in Canada and worldwide [1,2]. This fungal pathogen has been severely affecting cereal production in Canada since the early 1990s [3], causing significant yield and economic losses. In 2016, which was an epidemic year of FHB, ~$1 billion in losses were estimated in Canada primarily due to reduced crop yields and mycotoxin contamination of grains [4]. These losses to the wheat industry across North America far exceeded $2 billion [5]. Within Canada, there is a spatial distribution of *F. graminearum* chemotypes, with 3-acetyldeoxynivalenol (3ADON) dominating in Western and Atlantic Canada while having a low frequency of occurrence in Eastern Canada, including Ontario and Quebec [6].

During FHB infection, *F. graminearum* produces toxic secondary metabolites, known as mycotoxins, including trichothecene (type B) mycotoxins such as deoxynivalenol (DON) and its derivatives 15-acetyl-deoxynivalenol (15ADON) and 3ADON. Type B trichothecenes are characterized by the presence of a keto group at carbon atom 8 (C-8), compared to type A trichothecenes, where a keto group is absent [7]. In the early 2010s, a novel type A trichothecene, known as 3ANX (3-acetyl-NX; 7-α hydroxy,15-deacetylcalonectrin; NX-2), was identified among the secondary metabolites of *F. graminearum* isolate 02-264, isolated from non-agricultural grasses in Minnesota, USA [8]. Later, this isolate was genotyped as a 3ADON producer; however, chemical analysis indicated that none of the type B trichothecenes, including DON, 3ADON, 15ADON, and nivalenol (NIV), were produced [7,8,9]. Other nomenclature used in the literature for 3ANX and NX include NX-2 and NX-3 [10]. To date, a total of 189 isolates producing 3ANX have been identified worldwide, specifically in North America (Appendix A). Liang et al. [8] found 13 *F. graminearum* isolates that produced 3ANX from FHB-infected wheat heads collected between 1999 and 2000, 2006 and 2007, and 2011 and 2013 from Minnesota (MN), North Dakota (ND), and South Dakota (SD). These isolates were confirmed as 3ANX producers by mass spectrometric analysis. Kelly et al. [6] were the first to identify a 3ANX isolate in Canada upon evaluation of 728 *F. graminearum* isolates collected between 2005 and 2007 from Canada, among which 12 3ANX genotypes were detected. These 3ANX-positive isolates were also confirmed for toxin production in vitro using mass spectrometric methods. Interestingly, these 12 isolates that produce 3ANX were found to be distributed across Western, Eastern, and Atlantic Canada, and all of them in wheat, with 4 in Québec, 5 in Ontario, 1 in Prince Edward Island, and 2 in Saskatchewan. Later, Kelly et al. [11] characterized the spatial distribution of 3ANX isolates from a collection of 2515 *F. graminearum* isolates from 19 different countries. In this study, isolates producing 3ANX were identified in Canada (Québec, with 5 in barley; Manitoba, with 1 in oat) and the USA (Connecticut, with 2 in corn; Minnesota, with 1 in wheat). In addition, six new 3ANX producers isolated in North Dakota by Liang et al. [8] were also described in Kelly et al. [11]. Lofgren et al. [12] identified 56 strains via genetic testing in Willsboro, New York, from wheat, maize, and air samples, while Fulcher et al. [13] identified 24 strains via genetic testing in New York from wild grass spikes. Crippin et al. [14] identified seventeen 3ANX producers with the 15ADON genotype in Ontario, Canada, which co-produced noticeable levels of both 15ADON and 3ANX. Among those isolates was DAOM 242077, a positive 3ANX producer, isolated from winter wheat in Nova Scotia, Canada, in 2011. This isolate was used as a positive control in the current study. In addition, twenty-one 3ANX producers with the 15ADON genotype were identified from Ontario, Canada, and three 3ANX producers with the 15ADON genotype and three 3ANX producers with the 3ADON genotype were identified in Quebec, Canada [15]. Most recently, twenty-two 3ANX producers with the 15ADON genotype were identified in Ontario, Canada [16].

Additionally, NX is another novel mycotoxin, and 3ANX is the acetylated form of NX. The toxin 3ANX is similar to 3ADON without the keto group at C-8 in the same way that NX is similar to DON [17,18]. Similar to the deacetylation of 3ADON to DON, it was observed that 3ANX could be deacetylated to NX [7]. The toxicity of DON is well-documented in the literature, unlike these novel NX toxins. However, a recent intestinal toxicity study using histopathology, cytotoxicity, and transcriptome analyses [19] ranked the toxicity of NX, 3ANX, and DON in the following decreasing order: NX > DON ≫ 3ANX. In another toxicity study [7], NX inhibited protein biosynthesis similar to DON, while 3ANX was similar to 3ADON.

The spread of emerging *Fusarium graminearum* populations and NX mycotoxins poses significant risks to agriculture, food supply, and economic stability in Canada and globally. Therefore, it is important to assess the pathogen’s capacity to produce NX toxins and use the new knowledge to develop resistant wheat cultivars, as well as implement effective risk assessment and management strategies.

In this current study, we report the first large identification of 3ANX- and NX-producing isolates of *F. graminearum* in Western Canada among the 15ADON and 3ADON genotypes using a high-efficiency PCR assay and link genotypes to chemical analyses of the 3ANX metabolite.

## 2. Results

### 2.1. Fusarium graminearum on Wheat Samples

A total of 31,500 wheat spikes were screened for pathogen isolation and identification in the laboratory, where a subset of 6300 kernels from over 6 years were evaluated for pathogen identification. The results from kernels plated on PDA (25% strength) + streptomycin media showed that *F. graminearum* was the most frequently isolated pathogen species, accounting for 72.87% (2015), 98.3% (2016), 71.9% (2017), 74.4% (2018), 92.6% (2019), and 66.1% (2020) of isolations (Appendix A). A total of 382 monosporic pure cultures were prepared, among which 303 *F. graminearum* isolates were identified via morphological examination using a microscope and confirmed by sequencing the ribosomal intergenic spacer (IGS).

### 2.2. Mycotoxin Profiling Using High-Resolution Mass Spectrometry

The *F. graminearum* 3ANX positive control (DAOM 242077) strain produced DON, 3ANX, and NX. The isolate accumulated a significantly higher concentration of 3ANX at 1250 ppm (ppm: parts per million or mg/kg; ppb: parts per billion or µg/kg; Table 1), while the DON concentration was significantly lower (0.047 ppm). Under the in vitro growth conditions used in this study using rice media, 73 *F. graminearum* isolates that tested positive or negative for the NX-*Tri1*-F/R assay (Table 1, Appendix A) produced detectable levels of at least one of the mycotoxins: 15ADON, 3ADON, DON, 3ANX, or NX (Table 1). As expected with *F. graminearum* isolates, DON was accumulated in cultures of 96% of isolates from 2015 to 2020 with the exception of the HSW-16-14, HSW-16-19, and HSW-16-36 strains in 2016 (Figure 1, Table 1). The percent positive for 3ANX (63%) and 3ADON (66%) were similar across all years, suggesting all strains that accumulated the 3ANX toxin also co-produced 3ADON, except two strains (Figure 1 and Table 2). Most of the positive 3ANX isolates, as confirmed by mass spectrometry analysis, were located close to the USA border (Figure 2). An isolate of the 15ADON chemotype (HSW-15-23), as classified by Starkey (Table 1), also produced 3ADON and, interestingly, there was co-accumulation of 3ANX. A total of seventeen isolates showed a concentration of 3ANX higher than 1 ppm (concentrations range from 1.0 to 52 ppm). A significant correlation (r^2^ = 0.87) was observed between 3ADON and 3ANX among those positive strains (Figure 3). Similarly, a significant correlation (r^2^ = 0.89) was observed between DON and 3ANX (Figure 4).

Of the 73 *F. graminearum* isolates, only 36% (*n* = 26) were positive for 15ADON accumulation, and the concentrations ranged from 0.01 to 46.3 ppm (Figure 1 and Table 1 and Table 2). No correlation was observed between 15ADON and 3ANX or the NX toxin. Nivalenol accumulated in the cultures for 62% (*n* = 45) of isolates; however, the concentrations were lower than all the other mycotoxins and similar to the NX toxin. The concentrations ranged from 0.01 to 0.19 ppm with the exception of one isolate, HSW-15-21, collected in 2015, which accumulated 0.93 ppm of NIV (Table 1 and Table 2).

Among the *F. graminearum* strains analyzed from the year 2015, 63% (*n* = 21) accumulated the 3ANX toxin, with concentrations ranging between 0.044 and 5.42 ppm (Table 2). The ratios of 3ADON/3ANX concentrations ranged from 1.3 to 205 with a median value of 19.4, while the ratios of DON/3ANX ranged from 7.3 to 32.3 with a median value of 19.7 (Table 1). The isolates that coproduced 3ADON and 3ANX or DON and 3ANX accumulated less 3ANX compared to 3ADON or DON, respectively. The NX toxin was accumulated in 19% (*n* = 14) of samples among the 73 samples. Of these positive isolates, 13 were from 2015 and one was from 2020 (Table 1 and Table 2). The concentrations of NX toxins accumulated were lower compared to all the other mycotoxins and similar to the NIV toxin, with concentrations ranging from 0.07 to 0.62 ppm with the exception of one isolate, HSW-15-21, with a concentration of 3.22 ppm (Table 1 and Table 2). Five strains coproduced detectable levels of 3ADON, DON, 3ANX, and NX (HSW-15-11, HSW-15-21, HSW-15-39, HSW-15-7, and HSW-15-103). Five strains coproduced detectable levels of 15ADON, DON, and NX (HSW-15-27, HSW-15-31, HSW-15-57, HSW-15-85, and HSW-15-110). Three strains coproduced detectable levels of 15ADON, 3ADON, DON, 3ANX, and NX (HSW-15-23, HSW-15-87, and HSW-15-89). The isolate HSW-15-21 had the highest concentration of NX (3.2 ppm) among all isolates collected from the years 2015 to 2020.

Among the 14 *F. graminearum* isolates analyzed from the year 2016, 57.1% (*n* = 8) of them produced 3ANX. Eight isolates produced 3ADON, DON, and 3ANX, while three strains produced 15ADON and DON, and one strain produced only 15ADON (Table 1 and Table 2). The ratios of 3ADON/3ANX concentrations ranged from 7.7 to 11.1 with a median value of 10, while the ratios of DON/3ANX ranged from 6.9 to 27.3 with a median value of 18.5 (Table 1). Among the two *F. graminearum* isolates from the year 2017, 100% (*n* = 2) produced 3ANX along with 3ADON and DON. The mean ratio of 3ADON/3ANX concentrations was 8.8, while the mean ratio of DON/3ANX was 21.5 (Table 1). From the five *F. graminearum* isolates from the year 2018, 20% (*n* = 1) produced 3ANX (HSW-18-23). This isolate also coproduced 3ADON and DON. The remaining four isolates produced 15ADON and DON. The mean ratio of 3ADON/3ANX concentrations was 7.7, while the mean ratio of DON/3ANX was 5.3 (Table 1). Among the isolates from the year 2019, 40% (*n* = 2) of the five representative isolates produced 3ANX, 3ADON, and DON. From the remaining three isolates, two produced 15ADON and DON, while one (HSW-19-03) produced only DON. The mean ratio of 3ADON/3ANX concentrations was 9.0, while the mean ratio of DON/3ANX was 16.0 (Table 1). Among the isolates from the year 2020, 86% (*n* = 12) of the 14 representative isolates produced 3ANX and coproduced 3ADON and DON. Among them, the isolate HSW-20-01 produced the highest concentration of 3ANX (52 ppm). This was also the highest concentration of 3ANX among all isolates collected from 2015 to 2020. Among the remaining two isolates, one produced only 15ADON and the other produced only DON. The mean ratio of 3ADON/3ANX concentrations was 8.9, while the mean ratio of DON/3ANX was 18.8 (Table 1). The positive control 3ANX (DAOM 242077) isolate produced very high levels of 3ANX (1250 ppm) and NX (185.5 ppm) and very low levels of DON (0.047 ppm) in the rice medium.

### 2.3. Molecular Evaluation of Fusarium graminearum Isolates and Mycotoxin Analysis Confirmation

Trichothecene genotype analysis was conducted to examine the *Fusarium graminearum* chemotypes and 3ANX accumulation potential. Of the 303 *F. graminearum* isolates tested with the NX-*Tri1*-F/R primers, which tests for 3ANX-positive strains, 37 were found positive for the primers (Table 1, isolate IDs 1-37). Among these 37 positive isolates, mycotoxin analysis using high-resolution mass spectrometry confirmed 3ANX in 70.3% (*n* = 26). These data suggest that the NX-*Tri1*-F/R assay had a 70.3% efficiency in genotype matching the 3ANX metabolite. However, to confirm the efficiency of the NX-*Tri1*-F/R assay in genotype matching with the 3ANX metabolite, an additional 36 *F. graminearum* strains that were negative for the NX-*Tri1*-F/R primers (Table 1, isolate IDs 38-75, Appendix A) were tested for 3ANX by mass spectrometry. Interestingly, it was found that 55.6% (*n* = 20) of the isolates tested produced 3ANX, with concentrations ranging from 0.09 to 5.4 ppm (Table 1). In addition, 36% (*n* = 13) produced NX in the range of 0.07 to 3.2 ppm. Consequently, including the 3ANX positive control, DAOM 242077, (*n* = 74 strains), the assay efficiency dropped from 70.2% to 63.5% when genotype matching the 3ANX metabolite within the positive isolates. A total of 47 *F. graminearum* strains showed genotype consensus with the 3ANX metabolite data from mycotoxin analysis.

Twenty-four 15ADON genotypes and forty-five 3ADON genotypes, as identified using Starkey et al.’s [21] primers, accumulated 15ADON and 3ADON mycotoxins, respectively, and were confirmed using high-resolution mass spectrometry analysis. However, four *F. graminearum* isolates co-produced detectable levels of 3ADON and 15ADON (HSW-15-23, HSW-15-83, HSW-15-87, and HSW-15-89). The restriction fragment length polymorphism-polymerase chain reaction (RFLP-PCR) assays from both Liang et al. [8] and Toomajian [14] determined DAOM 242077 as a 3ANX genotype. This 3ANX positive control isolate produced 3ANX in the rice medium culture as confirmed by mass spectrometry analysis. However, the RFLP-PCR results were inconclusive in determining the genotype of any of the 73 *F. graminearum* strains analyzed (Table 1, Appendix A). The Toomajian assay determined 24 *F. graminearum* strains with genotype consensus against the 3ANX metabolite (Table 1, Appendix A), representing 33% consistency between the 3ANX genotype and metabolite analyses.

### 2.4. Sequencing of the Tri1 Flanking Regions Indicate Multiple Alleles Are Associated with NX

The *Tri1* gene and its flanking regions (3.3 kb) were amplified from the 73 *F. graminearum* isolates and the positive control, the DAOM 242077 isolate. The amplicon was sequenced and analyzed using multiple sequence alignment. The sequences were deposited in the GenBank (*accession numbers PQ587122 to PQ587195*). The resulting alignment was used for phylogenetic analysis. The phylogeny of the *Tri1* gene and its flanking regions resolved into multiple major clades (Figure 5). However, the grouping of *Tri1* genes does not correlate with the production of different mycotoxins. For example, 3ANX was produced by isolates that were distributed throughout the phylogenetic tree. Based on the sequencing results, the *Apo*I restriction site (GAATTC) in the fourth exon of the *Tri1* gene is present in all 73 *F. graminearum* isolates. The second *Apo*I restriction site (AAATTC), which was used to differentiate NX producers with the PCR-RFLP assay from Liang et al. [8], was absent in all tested isolates except the positive control, DAOM 242077 (Appendix A). The alignment of the region from the isolates in this study with 37 isolates that had published sequences of the *Tri1* gene (Appendix A) from Liang et al. [8] and Kelly et al. [6,11] produced similar results.

## 3. Discussion

In the present study, we performed the first large identification of 3ANX- and NX-producing isolates of *F. graminearum* in Western Canada. Also, we achieved the first identification of 15ADON chemotypes producing 3ANX in Western Canada and the first identification of 3ANX- and NX-producing isolates in Manitoba collected from wheat samples. To date, including this study, a total of 189 3ANX-producer isolates have been identified worldwide, specifically in North America [6,8,11,12,13,14,15,16]. These 3ANX-producing isolates have been isolated from wheat, barley, oat, corn, and wild grass. Most of the North American 3ANX isolates sharing 3ADON genotypes from the USA were identified in Minnesota, North Dakota, and South Dakota [6,8,11]. These USA states are located south of Manitoba, Canada, and most of the positive 3ANX (3ADON) isolates identified in this study were located close to the USA border. Gale et al. [23] and Liang et al. [8] showed that 3ADON-producing strains were limited to the Red River Valley defining the border between Minnesota and North Dakota in 1999–2000. Currently, there are very limited data on the spread of 3ANX and no data on NX-producing isolates in other parts of Canada, and future studies are warranted to investigate the expansion of these isolates to different geographical areas in Eastern and Western Canada.

The results from the current study indicate that the *F. graminearum* 3ADON genotype is now the most dominant of acetyl-DONs, i.e., among 3ADON and 15ADON. This is in agreement with previously reported trends on the 3ADON genotype in western Canada [1,8]. There has been a significant increase in the number and geographical expansion of 3ANX-producing isolates identified worldwide from 2002 to 2019 (Appendix A). About 60.5% of identified 3ANX producers in North America were collected from 2015 to 2019 (*n* = 66). A similar trend was observed in this study. In 2015, 63.6% of the isolates were 3ANX producers compared to 85.7% in 2020. The impact of environmental factors on NX isolates and their mycotoxin production remains largely unknown [24]. However, the increase in corn and soybean acreage, which could be alternate hosts in Manitoba, the deployment of moderately resistant wheat cultivars, and climate changes toward drought conditions could potentially be factors contributing to the increase in 3ANX-producing *F. graminearum* in recent years. In addition, the lack of testing methods for NX mycotoxins and the inconsistent performance of previous *Tri1* primers used in previous studies may also be factors.

The accumulation of 3ANX in the in vitro cultures of *F. graminearum* isolates reported by Crippin et al. [14] and Eli et al. [16] ranged from 0.029 to 3.8 ppm and 0.2 to 2.5 ppm, respectively. Similarly, the accumulation of 3ANX in the in vitro cultures of *F. culmorum* isolates ranged from 1.2 to 35.6 ppm [17]. In the in vitro cultures of *F. graminearum* isolates in the present study, higher concentrations of 3ANX were observed, ranging from 0.04 to 52 ppm. Notably, 37% of the isolates (*n* = 17) produced 3ANX concentrations exceeding 1.0 ppm.

It has been reported that the 3ANX toxin is typically deacetylated into NX in planta [7,25,26]; however, such deacetylation was incomplete in the in vitro cultures of *F. graminearum* isolates and, as a result, NX concentrations ranging from 0.07 to 3.2 ppm were observed. The 3ANX-positive isolate, DAOM 242077, also produced 185.5 ppm of the NX toxin under in vitro conditions. In addition, in contrast to reports that 3ANX-producing strains did not produce DON, NIV, or their acetylated derivatives [7,9], most of the isolates tested in the present study produced the 3ANX toxin simultaneously with DON, 3ADON, 15ADON, or NIV toxins. Similar findings were reported by Crippin et al. [14] and Eli et al. [16], where 3ANX-positive *F. graminearum* isolates simultaneously produced DON or 15ADON.

Among all the isolates that produced 3ANX, DON, and 3ADON, the ratios of 3ADON/3ANX were lower compared to the ratios of DON/3ANX, except in five isolates. No significance was observed among these five isolates. The only reported data on these ratios in the literature were on 15ADON/3ANX, which were lower in kernels than stalks in maize [16]. Further significance of these ratios could not be determined and warrants future studies. In Ontario, Canada, 80% of 15ADON isolates were able to produce 3ANX [14,15,25]; however, in Manitoba, Canada, where 3ADON isolates dominate, only 15% (*n* = 4) of the 15ADON isolates tested were shown to produce 3ANX in the present study. In Manitoba, 96% of 3ADON isolates were able to produce 3ANX.

To date, all assays primarily used to detect 3ANX strains were developed based on the *Tri1* gene [8,14,27,28]. In this research, assays from Crippin et al. [14] and Liang et al. [8] were tested against the PCR assay described in the present study. Based on the set of *F. graminearum* isolates tested in this study, the best assay to detect 3ANX-producing strains was developed in this study with a 63.5% overlap between the 3ANX genotype and metabolite data, as confirmed by mass spectrometry analysis. The next best assay was that of Crippin et al. [14] with only 33% overlap. Sequencing analysis and phylogenetic analysis of the *Tri1* gene and its flanking regions, as shown in Figure 5 and Appendix A, explained the discrepancy with the assays from Crippin et al. [14] and Liang et al. [8].

3ANX was produced by isolates that were distributed throughout the phylogenetic clades, and the grouping of *Tri1* genes did not correlate with the production of 3ANX (Figure 5). Our results were different from the recently published study by Gao et al. [27], where their *Tri1* gene phylogeny resolved type B trichothecene, NX, and T-2 strains into three different major clades. In addition, the *Tri1* gene’s second *Apo*I restriction site (AAATTC), which was used to discriminate NX producers with the PCR-RFLP [8], was notably absent in all the tested isolates except DAOM 242077 (Appendix A). Therefore, our results indicate that *Apo*I-based prediction of NX toxin producers is not reliable. Our results demonstrated that the existing genomic markers do not provide a complete representation of the 3ANX potential of *F. graminearum* strains. In the present research, no allelic variants of *Tri1* were found that could be directly correlated to the biosynthesis of 3ANX and NX toxins. This implies that trichothecene biosynthesis may not have a linear pathway in terms of substrate conversion to a single end product. Our results instead support that many different biosynthetic products are possible. It remains unclear as to the mechanism through which multiple products are being made. There is a wealth of knowledge supporting the idea that specific alleles of the biosynthetic pathway give rise to certain end products, including the gain, loss, and diversification of these genes, as well as some steps being non-enzymatic [29,30,31]. For example, the systematic disruption of biosynthesis genes results in the accumulation of different trichothecene intermediates; the disruption of *Tri1* results in a shift in the pathway toward calonectrin and 3-deacetylcalonectrin [31]. Likewise, the disruption of *Tri3*, a 15-acetyltransferase, reduces the efficiency of certain cyclization steps and results in the accumulation of unusual intermediates [31]. Therefore, it is possible that these pathways may not be as linear as once thought to be, and multiple products are possible as a result of the interactions between multiple enzymes and alleles that vary in their expression, kinetics, efficiencies, and substrate specificities, which may be compounded by variations in substrate abundance [31]. This is certainly an area that warrants further research. Consequently, it is possible that a large population of *F. graminearum* 3ANX-producers is distributed worldwide, and there is a need to re-examine *F. graminearum* culture collections to identify these 3ANX-producers given that chemotype assignment within the species may be from genetic testing, which may be inconsistent with the toxins being produced if the toxins were measured directly. Our results also challenge the notion of chemotypes, where isolates are assigned to a single toxin type since multiple different trichothecenes may be observed from a single isolate.

## 4. Conclusions

This study marks the first large identification of NX-producing isolates of *F. graminearum* in Western Canada. Most of the isolates that produced the 3ANX toxin also coproduced DON, 3ADON, 15ADON, or NIV. The currently available genomic markers, which were primarily developed based on the *Tri1* gene, were inefficient in determining 3ANX-producing *F. graminearum* isolates. Determining the pathogen’s ability to produce 3ANX and NX toxins under changing climates and screen cereal grain varieties for their resistance to NX-producing isolates will have a greater impact on food safety and security, reducing agriculture waste and improving economic gains in agriculture. Further studies are warranted to identify additional pathways for the biosynthesis of 3ANX, as well as for developing molecular diagnostic tools for the surveillance of field-collected samples. Analytical methods to detect novel NX-producing isolates are critical for all future studies in this field.

## 5. Materials and Methods

### 5.1. Fungal Isolation and Identification

A total of 630 wheat-producing farms were evaluated in Manitoba for fusarium head blight (FHB) from 2015 to 2020, with sampling periods between late July and early August of each year when most of the crops were at a growth stage of ZGS 73-85. Wheat crops cover almost 98% of cultivated acreage in the province of Manitoba (https://www.masc.mb.ca/masc.nsf/mmpp_index.html; accessed on 09 January 2025). Five wheat spikes (main stems) at 10 sites along a W-pattern in the field were sampled while avoiding sampling tillers. Kernels were surface-sterilized in a laminar flow bench for 1 min in 6% sodium hypochlorite, followed by 1 min in sterile water, and placed on a potato dextrose agar (PDA, 25% strength; Difco^TM^ 213400) amended with 0.02% streptomycin sulfate. Kernels on the PDA were incubated at 21–22 °C for 7 days under fluorescent light. The identification of *Fusarium* species was carried out via microscopic examination and morphological characterization using the criteria established by Leslie and Summerell [32]. The 3ANX-producing reference isolate, *F. graminearum* DAOM 242077, used as a positive control was obtained from the fungal culture collection of Agriculture and Agri-Food Canada, Ottawa, Canada, and the Canadian Grain Commission, Winnipeg, Canada.

### 5.2. DNA Extraction and Species Confirmation by Sequencing

Single-spore cultures from each field were plated onto a 60 mm quarter PDA (25% strength) amended with 0.02% streptomycin sulfate and incubated at 21–22 °C for 7 days under fluorescent light. DNA from the fungal mycelia on PDA was extracted using the protocol as described in Henriquez et al. [33] with the following modifications. Briefly, the mycelia were macerated in the presence of the TES extraction buffer (800 µL) containing 0.2 M Tris at pH 7.5, 10 mM EDTA at pH 8.0, and 0.5 M NaCl for 1 min at 1500 rpm using the 1500 MiniG^®^—Tissue Homogenizer and Cell Lyser (Spex SamplePrep, Metuchen, NJ, USA). Then, 7.5M ammonium acetate (400 µL) was added, and an aliquot of the supernatant (1000 µL) was transferred to a 2 mL microcentrifuge tube followed by the addition of chloroform-isoamyl alcohol (24:1, 500 μL). Approximately 600 µL of the top layer containing DNA was transferred to a new microcentrifuge tube. The chloroform-isoamyl alcohol step was repeated once more on the same extract to isolate the remaining DNA and transferred to the same tube. The DNA was precipitated from the solution by adding ice-cold isopropanol up to the 2.0 mL mark in the microcentrifuge tube, followed by vortexing and incubating at −20 °C for a minimum of 2 h. DNA washing, re-suspension, RNase A treatments, and quantification were followed as described in Henriquez et al. [33].

Species identity was confirmed by sequencing the ribosomal intergenic spacer (IGS), using the primers CNS1 and CNL12 [34]. The PCR reactions were performed in a 25 µL solution containing 1× Thermo Scientific DreamTaq Buffer (Thermo Fisher Scientific, Waltham, MA, USA), 1.2 mM MgCl_2_, 0.2 mM dNTPs, 0.16 μM of each of the primers, 0.8 mg/mL BSA, 1 unit of Thermo Scientific DreamTaq (Thermo Fisher Scientific, Waltham, MA, USA), and 40 ng of genomic DNA. Amplifications were performed in a C-1000 Thermal Cycler (Bio-Rad Laboratories, Hercules, CA, USA) programmed for an initial step of DNA denaturation at 95 °C for 3 min, followed by 35 cycles of 30 s at 95 °C, 30 s at 55 °C, and 1 min at 72 °C, and a final extension of 5 min at 72 °C. The PCR product was electrophoresed on 1.5% agarose gel, and the fragment size was estimated using a 1 kb Plus DNA Ladder (Thermo Fisher Scientific, Waltham, MA, USA). The PCR products were sequenced by the Genome Quebec Innovation Centre (Montreal, QC, Canada). Sequence homology was compared using BLAST analysis in the GenBank (blast.ncbi.nlm.nih.gov/Blast.cgi; accessed on 13 September 2024).

The fungal isolate cultures investigated in the present study were deposited into M.A. Henriquez’s culture collection at the Morden Research and Development Centre, Agriculture and Agri-Food Canada, Morden, MB, Canada (HSW-year-culture No.) and are available for research purposes upon written request.

### 5.3. F. graminearum Species-Specific PCR

*Fusarium graminearum* species-specific PCR was performed using the primers Fg16F/Fg16R, which produces a 410 bp product [20]. The PCR reactions were performed in 12.5 µL containing 1× Thermo Scientific DreamTaq Buffer (Thermo Fisher Scientific, Waltham, MA, USA), 1.2 mM MgCl_2_, 0.2 mM dNTPs, 0.16 μM of each of the primers, 0.8 mg/mL BSA, 0.625 units of Thermo Scientific DreamTaq (Thermo Fisher Scientific, Waltham, MA, USA), and 20 ng of genomic DNA. Amplifications were performed in a C-1000 Thermal Cycler (Bio-Rad Laboratories, Hercules, CA, USA) programmed for an initial step of DNA denaturation at 95 °C for 3 min, followed by 35 cycles of 30 s at 95 °C, 30 s at 55 °C, and 1 min at 72 °C, and a final extension of 5 min at 72 °C. The PCR products were electrophoresed on 1.5% agarose gels, and the fragment size was estimated using a 1 kb Plus DNA Ladder (Thermo Fisher Scientific, Waltham, MA, USA).

### 5.4. Trichothecene Genotype Identification

Multiplex PCR primers included the primers 12CON (5′-CATGAGCATGGTGATGTC-3′), 12NF (5′-TCTCCTCGTTGTATCTG G-30′), 12-15F (5′-TACAGCGGTCGCAACTTC-3′), and 12-3F (5′-CTTTGGCAAGCCCGTGCA-3′) [21], with expected products of 840 bp for NIV, 670 bp for 15ADON, and 410 bp for 3ADON chemotype. The PCR reactions were performed in 12.5 µL containing 1× Thermo Scientific DreamTaq Buffer (Thermo Fisher Scientific, Waltham, MA, USA), 2.0 mM MgCl_2_, 0.2 mM dNTPs, 0.2 μM of each of the primers, 1.6 mg/mL BSA, 0.5 U of Thermo Scientific DreamTaq (Thermo Fisher Scientific, Waltham, MA, USA) and 20 ng of genomic DNA. Amplifications were performed in a C-1000 Thermal Cycler (Bio-Rad Laboratories, Hercules, CA, USA) programmed as described above for Fg16F/Fg16R primers. In order to identify 3ANX isolates, we evaluated the PCR-RFLP assay using the forward primer *Tri1*F (5′-ATGGCTCTCATCACCAG-3′) and reverse primer *Tri1*R (5′-CAATTCCAATCGCAGACAA-3′) and digestion using the endonuclease *Apo*I [8]. In order to differentiate between ADON and 3ANX producers, we evaluated the Toomajian assay [14] following their PCR conditions. The PCR assay used a common reverse primer (*TRI1*-R; 5′-TTCCTGCAGGGGCTTGATG-3′) and differing forward primers for the 3ADON (5′-AATGCTCGCGAACTAATCAC-3′) and 3ANX (5′-AATGCTAGCGAAATGATCAA-3′). In addition, the *TRI1* allele-specific primers NX-*Tri1*-F (5′-TCGATGTTAATTGTTTTTGTGTA-3′) and NX-*Tri1*-R (5′-AGCCAGCTGGGTTTCTTG-3′) designed from the cytochrome P450 monooxygenase (*TRI1*) gene (accession: KX183557.1) were used in the current study. The PCR reactions for this set of primers were performed in 20 µL containing 1× Thermo Scientific DreamTaq Buffer (Thermo Fisher Scientific, Waltham, MA, USA), 1.5 mM MgCl_2_, 0.2 mM dNTPs, 0.16 μM of each of the primers, 0.8 mg/mL BSA, 1 U of Thermo Scientific DreamTaq (Thermo Fisher Scientific, Waltham, MA, USA), and 50 ng of genomic DNA. Amplifications were performed in a C-1000 Thermal Cycler (Bio-Rad Laboratories, Hercules, CA, USA) programmed for an initial step of DNA denaturation at 94 °C for 2 min, followed by 35 cycles of 30 s at 94 °C, 30 s at 56.3 °C, and 20 s at 72 °C, and a final extension of 10 min at 72 °C. The PCR product (130 bp) was electrophoresed on 1.5% agarose gels, and the fragment size was estimated using a 1 kb Plus DNA Ladder (Thermo Fisher Scientific, Waltham, MA, USA).

### 5.5. Culturing of Fungi in Rice Medium

The *Fusarium graminearum* NX-2 strains that were positive and negative for the molecular primers and preserved at −80 °C (filter paper Whatman No.1) were plated onto Petri dishes (100 mm) containing Spezieller–Nährstoffar Agar (SNA) and incubated at 22 °C for 10 days under a combination of fluorescent UV lights [35]. Rice cultures were prepared in 50 mL glass beakers wrapped with aluminum foil by autoclaving 10 g of washed rice (Organic long-grain white rice from a supermarket) with distilled water. Rice from the same batch of organic long-grain white rice was used for all experiments. The sterilized rice in the beakers was inoculated with 1 mL of diluted inoculum at a concentration of 5 × 10^4^ macroconidia/mL or water (control). The cultures were incubated in replicates (*n* = 3 to 5) for 10 days at 22 °C and freeze-dried for 48 h before extraction for fungal secondary metabolites. All biological replicates per strain were extracted separately for fungal secondary metabolites.

### 5.6. Chemicals and Reagents for Fungal Secondary Metabolite Extraction

Acetonitrile, methanol, formic acid (Optima grade, Fisher Scientific, Ottawa, ON, Canada), and Milli-Q water (18.2 MΩ.cm, total organic carbon < 3.00 ppb, IQ 7000, Millipore, Bedford, MA, USA) were used for solvents, sample extraction, and the preparation of analytical and calibration standards. 3ANX (7-α hydroxy, 15-deacetylcalonectrin) and NX (7α hydroxy, 3, 15-dideacetylcalonectrin) toxins were provided by Agriculture and Agri-Food Canada, London, Canada, and Carleton University, Ottawa, Canada. High-purity (>95%) analytical standards of deoxynivalenol (DON), 15-acetyl deoxynivalenol (15ADON), 3-acetyl deoxynivalenol (3ADON), and nivalenol (NIV) were purchased from Sigma-Aldrich (currently MilliporeSigma Canada Ltd., Etobicoke, ON, Canada). Stable-isotope 13C-labeled internal standards including ^13^C_15_-DON, ^13^C_17_-15ADON, ^13^C_17_-3ADON, and ^13^C_15_-NIV were purchased from Romer (Vancouver, BC, Canada). All stock solutions were prepared in acetonitrile while working standards of mycotoxin mixtures were prepared in methanol and preserved at −20 °C.

### 5.7. Sample Extraction for Fungal Secondary Metabolites

The rice samples were ground to a fine particle size using a mortar and pestle. The mortar and pestle were thoroughly cleaned between samples to prevent cross-contamination. Subsequently, a sub-sample (1 g) was weighed into a clean 10 mL centrifuge tube, followed by the addition of a solvent mixture (10 mL) containing acetonitrile (75% *v*/*v*), methanol (10% *v*/*v*), and water (15 *v*/*v*). Secondary metabolite extraction was carried out using previously established methods [36,37] with minor modifications as described below. All tubes containing sample–solvent mixtures were thoroughly mixed by inverting 5 times to thoroughly wet all of the sample material, followed by sonication (30 min) at 23 °C. The tubes were then transferred onto a rotatory shaker for further extraction at 40 rpm (30 min). After the extraction, the sample–solvent mixture was centrifuged at 5000× *g* for 30 min at 4 °C to separate the supernatant. The supernatant was transferred into a clean tube (10 mL flat-bottomed) using a syringe filter (0.2 µm nylon, 15 mm, Phenomenex, CA, USA) fitted to a 10 mL syringe. An aliquot (5 mL) of the supernatant was transferred into a clean 40 mL amber glass vial for concentration and evaporation. The evaporation was achieved using a sample evaporator (Rocket Synergy 2, ThermoFisher, Mississauga, ON, Canada). The 40 mL glass vials containing the extract were placed inside the evaporator and vacuum-dried to completion using centrifugation at 50 °C for 45 min. The evaporated extracts were resuspended in 1 mL of a methanol/water (50:50, *v*/*v*) solution containing both 0.1% formic acid and 5 mM ammonium formate via sonicating (5 s) and vortexing (5 s). The resuspended extract was transferred to a 2 mL amber liquid chromatography vial for mass spectrometric analysis. Uninoculated control samples of the rice medium were used for fortification and recovery experiments. Ground control rice samples (1 g, *n* = 6) were fortified with a mixture containing 3ANX, NX, DON, 3ADON, 15ADON, and NIV at 100 µg/kg. Control blanks and fortified samples were extracted simultaneously with each batch of samples for quality control as described earlier. The mean recoveries for all mycotoxins quantified are presented in Table 3.

### 5.8. Sample Analysis and Data Acquisition Using UHPLC-HRMS

All sample extracts were spiked with 10 µL of a stable isotope 13C-labeled internal standard mix containing 2.5 µg/mL each (^13^C_15_-DON, ^13^C_17_-3-acetyl DON, ^13^C_17_-15-acetyl DON, and ^13^C_15_ nivalenol) prior to analysis. All extracts were analyzed simultaneously for six trichothecene mycotoxins including 3ANX, NX, deoxynivalenol, 3-acetyl deoxynivalenol, 15-acetyl deoxynivalenol, and nivalenol. The analysis was conducted using ultra-high-performance liquid chromatography (UHPLC, Vanquish, ThermoFisher, Mississauga, ON, Canada) coupled with a high-resolution mass spectrometer (HRMS, Tribrid ID-X, ThermoFisher, Mississauga, ON, Canada). The samples were maintained at 4 °C during analysis. Analyte separation was achieved on a Kinetex F5 LC column maintained at 35 °C (1.7 µm, 100 Å, 100 × 2.1 mm, Phenomenex, CA, USA) using gradient elution (Table 4) with water (solvent A) and methanol (solvent B), containing both 0.1% formic acid and 5 mM ammonium formate as modifiers and with a total run time of 13 min. Heated-electrospray ionization (H-ESI-II) in positive ionization mode was used to achieve a steady-state electrospray.

Full-scan mass spectral data on the extracts were acquired using the HRMS instrument method operating in positive ionization mode with an orbitrap mass resolution of 120,000 for parent ions (*m*/*z*, mass-to-charge ratio) and a mass tolerance of <5 ppm. The fragmentation of parent ions was achieved via higher-energy collision-induced dissociation (HCD) with stepped collision energies (15, 20, and 25%) in the presence of high-purity nitrogen gas. Product ion data were acquired using an orbitrap mass resolution of 30,000. Parent and fragment ion *m*/*z* values were verified via the infusion of pure analytical standards (Table 3). The identification and confirmation of analytes was conducted using accurate mass (*m*/*z*) values of parent and fragment ions, while the quantification of analytes was carried out using parent ions. All sample data were acquired using Xcalibur ver.4.2, and data processing was carried out using TraceFinderTM v.4.1 and FreeStyle ver.1.6 (ThermoFisher Scientific, USA).

### 5.9. Sequencing Flanking Regions of the Tri1 Gene

Freeze-dried rice cultures were ground into a fine powder and stored at −20 °C until further use. Genomic DNA was extracted from the 73 different rice cultures (2 g) using the DNeasy maricon food kit (Qiagen 69514) according to the manufacturer’s instructions. The primers *Tri1*-NX-Flanking-F (CCGGATGCTTGACGTAGATT) and *Tri1*-NX-Flanking-R (TTTTTGGGCGAGATGTTACC) were designed to amplify 1.4 kb upstream of the *Tri1* gene start codon and 400 bp downstream of the *Tri1* gene stop codon. The flanking regions of *Tri1* (3.3 kb) were amplified using Q5 High-Fidelity polymerase (NEB M0491S) according to the manufacturer’s instructions. Amplicons were confirmed using 1.2% agarose gel electrophoresis, purified, and sequenced using Oxford Nanopore Technologies (Eurofins Genomics, Louisville, KY, USA). Sequences were aligned and analyzed using the CLC Genomics Workbench (ver. 24.0.1, Qiagen, Redwood City, CA, USA). The neighbor-joining phylogenetic tree of the *Tri1* gene and its flanking regions annotated with the mycotoxin (DON, 3ADON, 15ADON, and 3ANX) dataset was drawn using iTOL V6 [22].

### 5.10. Statistical Analysis

Fungal in vitro experiments and mycotoxin extractions were performed on 5 replicates, and mean concentrations were calculated. Pearson coefficients of correlation were calculated to measure the linear association between 3ANX and other mycotoxins, including DON, 3ADON, 15ADON, and NIV. All statistical analyses were performed using SigmaPlot software (ver. 15, Frankfurt, Germany). The limit of detection (LOD) and limit of quantification (LOQ) values for the mycotoxin extraction method were calculated using the measured observed mean concentrations and standard deviations of spiked control samples. The LOD was calculated as 3 times s’o, where s’o was calculated as the standard deviation divided by the square root of the number of replicates. The LOQ was calculated as 10 times s’o. The LOD and LOQ values are presented in Table 3.

## Figures and Tables

**Figure 1 toxins-17-00045-f001:**
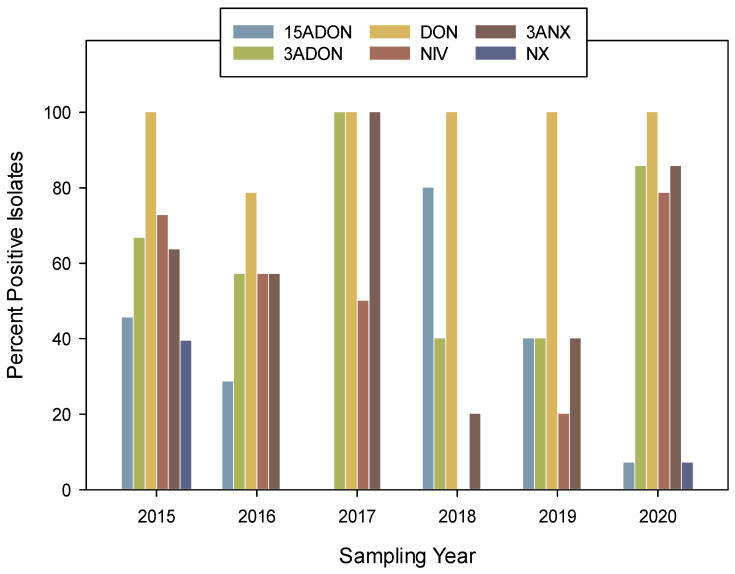
Total number of positive isolates (%) for various toxin accumulation among isolates collected from 2015 to 2020.

**Figure 2 toxins-17-00045-f002:**
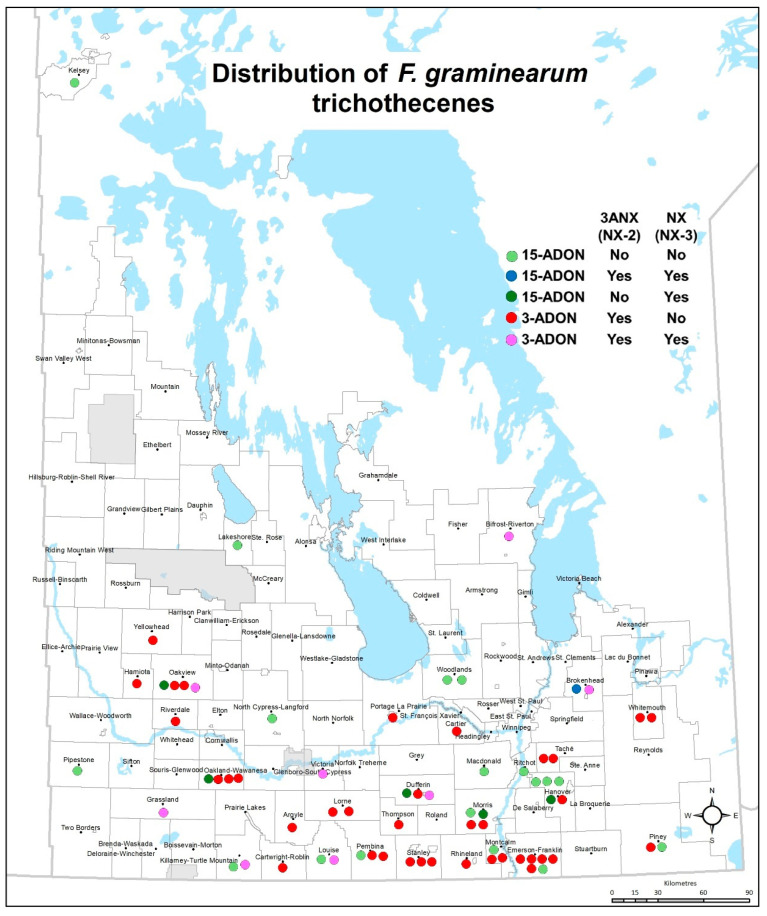
Distribution of *F. graminearum* isolates producing different trichothecene mycotoxins from across Manitoba, Canada.

**Figure 3 toxins-17-00045-f003:**
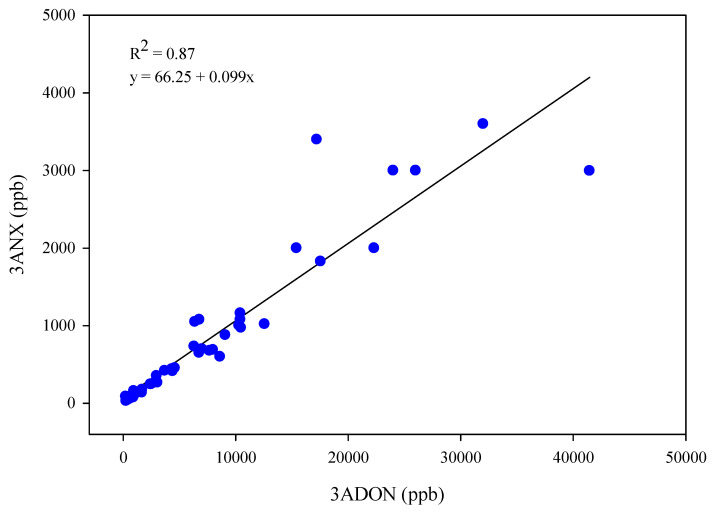
Relationship (Pearson correlation) between 3-acetyl DON (3ADON) and 3ANX. Each culture (Table 1) is represented by a single data point (blue dots). Outliers (HSW-15-11, HSW-15-21, HSW-15-103, and HSW-20-01) were removed.

**Figure 4 toxins-17-00045-f004:**
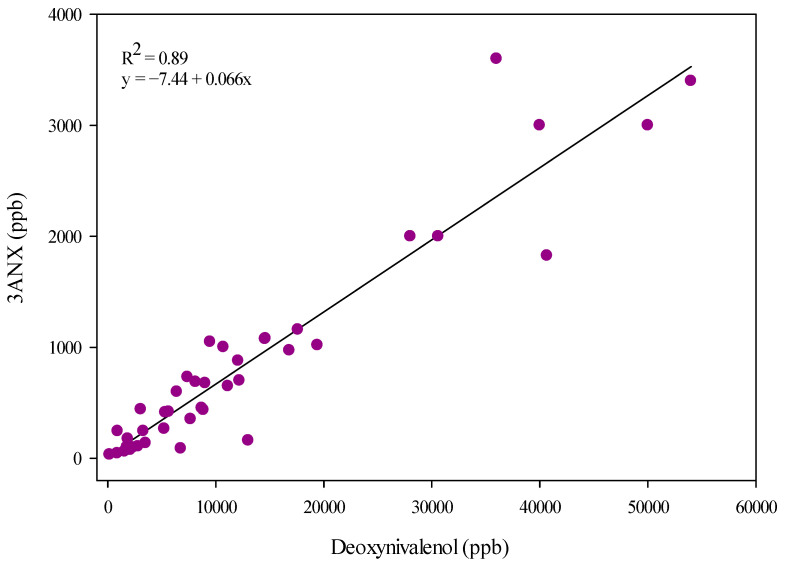
Relationship (Pearson correlation) between deoxynivalenol (DON) and 3ANX. Each culture (Table 1) is represented by a single data point (purple dots). Outliers (HSW-15-11, HSW-15-21, HSW-15-89, HSW-15-103, and HSW-20-01) were removed.

**Figure 5 toxins-17-00045-f005:**
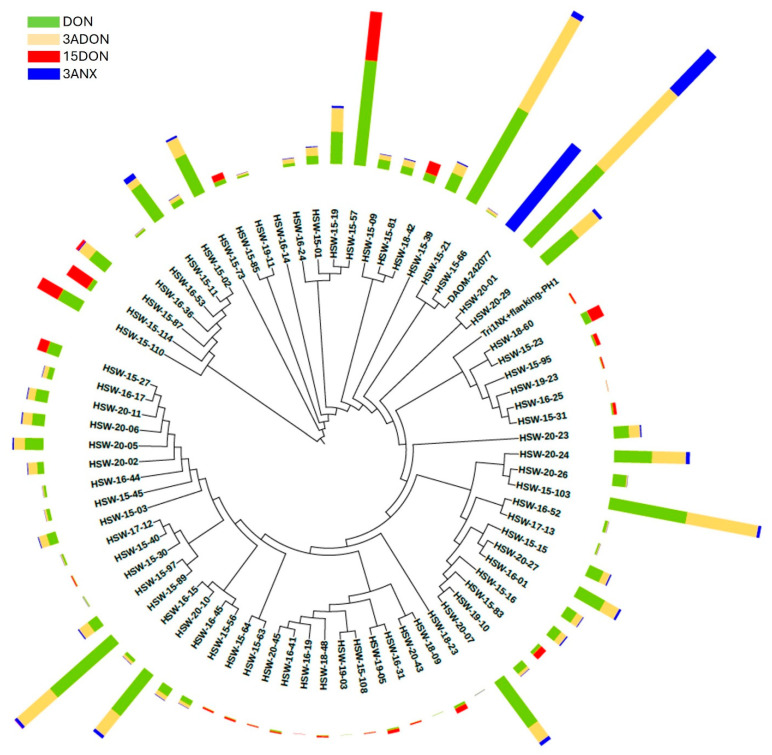
Neighbor-joining phylogenetic tree of the *Tri1* gene and its flanking regions (*n* = 74) annotated with mycotoxin (DON, 3ADON, 15ADON, and 3ANX) dataset. Multi-value bar graph depicts concentrations of DON (green), 3ADON (light orange), 15ADON (red), and 3ANX (blue) in ppb. The upper limit of mycotoxin concentration was set to 10,000 ppb. The tree was drawn using iTOL V6 [22].

**Table 1 toxins-17-00045-t001:** Mycotoxin production in the in vitro cultures of rice medium by various isolates evaluated for 3ANX positivity using five different primers from the literature and ratios of their concentrations.

ID	Collection No.	HSW ID	*F. graminearum* Marker (Nicholson ^1^)	Trichothecene Cluster Type (Starkey ^2^)	(Toomajian ^3^)		NX (Liang ^4^)	NX-*Tri1*-F/R ^5^	15ADON	3ADON	DON	3ANX	NX	NIV	Ratio 3ADON/3ANX	Ratio DON/3ANX
					ADON	ANX	*TRI1*		Mean (ppb ^6^)							
	3ANX+	DAOM 242077	+	3ADON	-	+	+	+	ND ^7^	ND	47	1,250,000	185,500	21	—	—
1	1	HSW-15-1	+	3ADON	+	+	−	+	ND	7978	8136	688	ND	42	11.6	11.8
2	2	HSW-15-2	+	3ADON	+	+	−	+	ND	3683	5647	420	ND	34	8.8	13.4
3	3	HSW-15-3	+	3ADON	+	+	−	+	ND	7656	9026	678	ND	38	11.3	13.3
4	9	HSW-15-9	+	3ADON	+	+	−	+	ND	4580	8697	453	ND	12	10.1	19.2
5	15	HSW-15-15	+	3ADON	+	+	−	+	ND	6763	14,543	1077	ND	76	6.3	13.5
6	16	HSW-15-16	+	3ADON	+	-	−	+	ND	6363	9470	1050	ND	28	6.1	9.0
7	30	HSW-15-30	+	3ADON	+	-	−	+	ND	410	890	44	ND	30	9.3	20.2
8	45	HSW-15-45	+	3ADON	+	+	-	+	ND	1657	3507	137	ND	39	12.1	25.5
9	56	HSW-15-56	+	3ADON	+	+	-	+	ND	3040	5227	267	ND	60	11.4	19.6
10	64	HSW-15-64	+	15ADON	+	+	-	+	1342	ND	531	ND	ND	ND	—	—
11	66	HSW-15-66	+	3ADON	+	+	-	+	ND	2420	920	246	ND	11	9.8	3.7
12	81	HSW-15-81	+	3ADON	+	+	-	+	ND	6303	7380	733	ND	57	8.6	10.1
13	83	HSW-15-83	+	15ADON	+	+	-	+	6210	48	3610	ND	ND	40	—	—
14	89	HSW-15-89	+	3ADON	+	+	-	+	12	41,447	77,547	2997	625	98	13.8	25.9
15	97	HSW-15-97	+	3ADON	+	+	-	+	ND	10,287	10,700	1003	ND	37	10.3	10.7
16	108	HSW-15-108	+	15ADON	+	+	-	+	537	ND	114	ND	ND	31	—	—
17	120	HSW-16-01	+	3ADON	+	+	-	+	ND	6743	11120	650	ND	10	10.4	17.1
18	134	HSW-16-15	+	3ADON	+	+	-	+	ND	2503	3300	247	ND	25	10.1	13.4
19	136	HSW-16-17	+	3ADON	+	+	-	+	ND	4387	5320	413	ND	14	10.6	12.9
20	133	HSW-16-14	+	3ADON	+	+	-	+	ND	ND	ND	ND	ND	16	—	—
21	138	HSW-16-19	+	15ADON	+	+	-	+	32	ND	ND	ND	ND	ND	—	—
22	143	HSW-16-24	+	3ADON	+	+	-	+	ND	4333	3053	442	ND	21	9.8	6.9
23	144	HSW-16-25	+	15ADON	+	+	-	+	90	ND	28	ND	ND	ND	—	—
24	155	HSW-16-36	+	3ADON	+	+	-	+	ND	ND	ND	ND	ND	ND	—	—
25	160	HSW-16-41	+	15ADON	+	+	-	+	1013	ND	1360	ND	ND	14	—	—
26	163	HSW-16-44	+	3ADON	+	+	-	+	ND	960	2333	97	ND	ND	9.9	24.1
27	164	HSW-16-45	+	3ADON	+	+	-	+	ND	4330	8857	437	ND	ND	9.9	20.3
28	171	HSW-16-52	+	3ADON	+	+	-	+	ND	837	2800	108	ND	21	7.7	25.9
29	172	HSW-16-53	+	3ADON	+	+	-	+	ND	900	2090	77	ND	14	11.7	27.3
30	192	HSW-17-12	+	3ADON	+	+	-	+	ND	777	1793	103	ND	ND	7.5	17.4
31	193	HSW-17-13	+	3ADON	+	+	-	+	ND	617	1560	61	ND	18	10.2	25.7
32	226	HSW-18-23	+	3ADON	+	+	-	+	ND	249	170	32	ND	ND	7.8	5.3
33	251	HSW-18-48	+	15ADON	+	+	-	+	1150	ND	737	ND	ND	ND	—	—
34	287	HSW-19-03	+	15ADON	+	+	-	+	ND	ND	17	ND	ND	ND	—	—
35	294	HSW-19-10	+	3ADON	+	+	-	+	ND	2967	7667	353	ND	26	8.4	21.7
36	295	HSW-19-11	+	3ADON	+	+	-	+	ND	1677	1847	176	ND	ND	9.5	10.5
37	363	HSW-20-45	+	15ADON	+	+	-	+	730	ND	243	ND	ND	ND	—	—
38	11	HSW-15-11	+	3ADON	+	-	-	-	ND	6867	38,000	5213	250	133	1.3	7.3
39	19	HSW-15-19	+	3ADON	+	-	-	-	ND	22,300	30,600	2000	ND	126	11.2	15.3
40	23	HSW-15-23	+	15ADON	+	-		-	12,333	216	6767	88	120	46	2.5	76.9
41	27	HSW-15-27	+	15ADON	+	-		-	9137	ND	12,497	ND	145	39	—	—
42	31	HSW-15-31	+	15ADON	+	+	-	-	2847	ND	1617	ND	72	ND	—	—
43	39	HSW-15-39	+	3ADON	+	-	-	-	ND	10,467	16,800	973	203	ND	10.8	17.3
44	40	HSW-15-40	+	15ADON	+	-		-	865	ND	649	ND	ND	ND	—	—
45	57	HSW-15-57	+	15ADON	+	-		-	46,300	ND	121,667	ND	308	137	—	—
46	63	HSW-15-63	+	15ADON	+	-		-	1478	ND	704	ND	ND	ND	—	—
48	73	HSW-15-73	+	3ADON	+	-	-	-	ND	17,533	40,667	1827	146	109	9.6	22.3
50	85	HSW-15-85	+	15ADON	+	+	-	-	6136	ND	4467	ND	110	ND	—	—
51	87	HSW-15-87	+	3ADON	+	-	-	-	2319	12,552	19,400	1020	232	128	12.3	19.0
52	95	HSW-15-95	+	15ADON	+	-	-	-	3633	ND	2020	ND	ND	ND	—	—
53	103	HSW-15-103	+	3ADON	+	-	-	-	ND	68,400	74,667	2667	261	55	25.7	28.0
54	110	HSW-15-110	+	15ADON	+	-	-	-	22,200	ND	23,200	ND	350	ND	—	—
55	114	HSW-15-114	+	15ADON	+	+	-	-	23,400	ND	4647	ND	ND	ND	—	—
56	150	HSW-16-31	+	15ADON	+	+	-	-	1020	ND	320	ND	ND	ND	—	—
57	212	HSW-18-09	+	15ADON	+	-	-	-	4800	ND	2600	ND	ND	ND	—	—
58	245	HSW-18-42	+	15ADON	+	+	-	-	11,400	ND	7600	ND	ND	ND	—	—
59	263	HSW-18-60	+	15ADON	+	-	-	-	1400	ND	460	ND	ND	ND	—	—
60	289	HSW-19-05	+	15ADON	+	-	-	-	3000	ND	1460	ND	ND	ND	—	—
61	307	HSW-19-23	+	15ADON	+	-	-	-	1180	ND	820	ND	ND	ND	—	—
62	319	HSW-20-01	+	3ADON	+	-	-	-	ND	500,000	188,000	52,000	ND	194	9.6	3.6
63	320	HSW-20-02	+	3ADON	+	-	-	-	ND	8600	6400	600	ND	ND	14.3	10.7
64	323	HSW-20-05	+	3ADON	+	-	-	-	ND	10,400	17,600	1160	ND	50	9.0	15.2
65	324	HSW-20-06	+	3ADON	+	-	-	-	ND	9067	12,067	880	240	48	10.3	13.7
66	325	HSW-20-07	+	3ADON	+	-	-	-	ND	17,200	54,000	3400	ND	60	5.1	15.9
67	328	HSW-20-10	+	3ADON	+	-	-	-	ND	24,000	50,000	3000	ND	82	8.0	16.7
68	329	HSW-20-11	+	3ADON	+	-	-	-	ND	7000	12,200	700	ND	28	10.0	17.4
69	341	HSW-20-23	+	3ADON	+	-	-	-	ND	10,400	14,600	1080	ND	50	9.6	13.5
70	342	HSW-20-24	+	3ADON	+	-	-	-	ND	32,000	36,000	3600	ND	76	8.9	10.0
71	344	HSW-20-26	+	3ADON	+	-	-	-	ND	940	13,000	160	ND	18	5.9	81.3
72	345	HSW-20-27	+	3ADON	+	-	-	-	ND	15,400	28,000	2000	ND	66	7.7	14.0
73	347	HSW-20-29	+	3ADON	+	-	-	-	ND	26,000	40,000	3000	ND	96	8.7	13.3
74	361	HSW-20-43	+	15ADON	+	-	-	-	ND	ND	116	ND	ND	ND	—	—
75 ^8^	21	HSW-15-21	+	3ADON				-	ND	1,111,050	175,040	5420	3224	925	205.0	32.3

^1^ Nicholson et al., 1998 [20]. ^2^ Starkey et al., 2007 [21]. ^3^ Crippin et al., 2019 [14]. ^4^ Liang et al., 2014 [8]. ^5^ Primers from the present research. ^6^ ppb: parts per billion or µg/kg. ^7^ ND: Not detected. ^8^ Total number of isolates in the table is 73 plus one 3ANX-positive DAOM 242077 isolate.

**Table 2 toxins-17-00045-t002:** Total number of positive isolates for various mycotoxins and concentration ranges detected in the samples.

Sampling Year		2015	2016	2017	2018	2019	2020
Mycotoxins	Total number of isolates (73)	33	14	2	5	5	14
**3ANX**	3ANX + (46)	21	8	2	1	2	12
	Number of +ve (%) ^1^	64	57	100	20	40	86
	Min (ppm)	0.04	0.08	0.06	0.03	0.18	0.16
	Max (ppm)	5.42	0.65	0.10	0.03	0.35	52.0
	**Total percent of +ve isolates (%)**	**63**					
**NX**	NX + (14)	13	0	0	0	0	1
	Number of +ve (%)	39	0	0	0	0	7
	Min (ppm)	0.07	0	0	0	0	0.24
	Max (ppm)	3.22	0	0	0	0	0.24
	**Total percent of +ve isolates (%)**	**19**					
**3-acetyl** **deoxynivalenol (3ADON)**	3ADON + (48)	22	8	2	2	2	12
	Number of +ve (%)	67	57	100	40	40	86
	Min (ppm)	0.05	0.84	0.62	0.25	1.68	0.94
	Max (ppm)	1111	6.74	0.78	0.25	2.97	500
	**Total percent of +ve isolates (%)**	**66**					
**15-acetyl** **deoxynivalenol (15ADON)**	15ADON + (26)	15	4	0	4	2	1
	Number of +ve (%)	46	29	0	80	40	7
	Min (ppm)	0.01	0.03	0.0	1.15	1.18	0.73
	Max (ppm)	46.3	1.02	0.0	11.4	3.0	0.73
	**Total percent of +ve isolates (%)**	**36**					
**Deoxynivlenol (DON)**	DON + (70)	33	11	2	5	5	14
	Number of +ve (%)	100	79	100	100	100	100
	Min (ppm)	0.11	0.03	1.56	0.17	0.02	0.12
	Max (ppm)	175.0	11.1	1.79	7.60	7.67	188.0
	**Total percent of +ve isolates (%)**	**96**					
**Nivalenol (NIV)**	NIV + (45)	24	8	1	0	1	11
	Number of +ve (%)	73	57	50	0	20	79
	Min (ppm)	0.01	0.01	0.02	0.0	0.03	0.02
	Max (ppm)	0.93	0.02	0.02	0.0	0.03	0.19
	**Total percent of +ve isolates (%)**	**62**					

^1^ Decimals were rounded to the nearest number for percentages.

**Table 3 toxins-17-00045-t003:** Mycotoxins, mass-to-charge ratios (*m*/*z*) of parent and fragment ions, retention times, recoveries, limits of detection (LODs), and limits of quantification (LOQs) used in the analytical method.

Mycotoxin	Parent Ion, [M+H]^+^ (*m*/*z*)	Fragment Ions (*m*/*z*)	Retention Time (min)	Recoveries (%) Mean ± SD	LOD (ppb)	LOQ (ppb)
3ANX (NX-2)	325.1643	199.1116 171.1166 121.0645 183.1170	8.29	81 ± 7	5.9	19.5
NX (NX-3)	283.1540 [M+H]^+^ 265.1434 [M-H_2_O+H]^+^	265.1434 217.1229 247.1333 199.1121	4.71	96 ± 7	5.9	19.9
DON	297.1333	261.1120 249.1119 203.1065	4.81	76 ± 6	0.7	2.4
^13^C_15_-DON	312.1841	-	4.81	-	-	-
15ADON	339.1438	231.1016 321.1332 261.1121	7.72	74 ± 4	0.2	0.6
^13^C_17_-15ADON	356.2014	-	7.72	-	-	-
3ADON	339.1438	231.1016 321.1332 261.1121 137.0592	7.91	81 ± 3	1.3	3.2
^13^C_17_-3ADON	356.2014	-	7.91	-	-	-
NIV	313.1282	247.0962 229.0858 205.0857	3.61	91 ± 6	4.1	13.5
^13^C_15_-NIV	328.1791	-	3.61	-	-	-

**Table 4 toxins-17-00045-t004:** Gradient elution method for trichothecene mycotoxins in positive ionization mode.

Flow Rate: 0.3 mL/min	
Time (min)	% B
0.00	5
4.00	20
4.50	35
8.50	35
11.00	100
11.05	5
13.00	5

## Data Availability

The original contributions presented in this study are included in the article/Appendix A. Further inquiries can be directed to the corresponding authors.

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
