# Peer review of "The First Large Identification of 3ANX and NX Producing Isolates of *Fusarium graminearum* in Manitoba, Western Canada"

_toxins, 2025, doi:10.3390/toxins17010045_

Round 1
Reviewer 1 Report
Comments and Suggestions for Authors
In discussion chapter, the author could give some point about the 3ANX biosynthesis pathway based on this study and some new papers about 15-Deacetylcalonectrin acetylation.
Author Response
Comment 1: In discussion chapter, the author could give some point about the 3ANX biosynthesis pathway based on this study and some new papers about 15-Deacetylcalonectrin acetylation.
Response 1: We thank the reviewer for this comment. We added additional points and new citation to address this comment [Page 16, lines 337-348]. Further elaboration on biosynthesis and pathways would need more information and data to support the arguments, which is beyond the scope of the current study. In this regard, further studies are underway which will help explain potential biosynthesis pathway.
Reviewer 2 Report
Comments and Suggestions for Authors
The authors systematically identified the 3ANX and NX producing isolates of fusarium graminearum, which is the major pathogen for fususarium head blight in wheat. This is a very important and interesting research for wheat breeders, wheat producers and many related researchers. Please revise the following points to improve the paper.
(1) Western Canada is a big area. Monitoba is part of Western Canada. You collected a total of 31,500 wheat spikes over a period of 6 years (2015-2020) within Manitoba. Please add Monitoba in the title of this paper.
(2) Please show the wheat growing regions of Moitoba in Figure 2.
(3) Is there any problem for Figure 1?
(4) Please explain “ppm” and “ppb” when first use.
(5) Line 447: What kind of rice did you used for preparing rice media? Did you buy it from supermarket? Did you used the same rice throughout the research? Do you the protein and starch contents of the rice used?
(6) Please provide the company name of PDA used.
Author Response
Comment 1: The authors systematically identified the 3ANX and NX producing isolates of fusarium graminearum, which is the major pathogen for fususarium head blight in wheat. This is a very important and interesting research for wheat breeders, wheat producers and many related researchers. Please revise the following points to improve the paper.
Response 1: Thank you and we greatly appreciate for recognizing the importance of this research work for wheat researchers and stakeholders. Thank you for providing valuable comments and suggestions which helped improve the manuscript. All the comments have been addressed as mentioned below.
Comment 2: Western Canada is a big area. Monitoba is part of Western Canada. You collected a total of 31,500 wheat spikes over a period of 6 years (2015-2020) within Manitoba. Please add Monitoba in the title of this paper.
Response 2: We agree with the reviewer. We revised the title to include Manitoba. Page 1 - Line 1-3.
Comment 3: Please show the wheat growing regions of Moitoba in Figure 2.
Response 3: Thanks for the comment. Wheat is grown in almost 98% of farmlands in the province. We included this new information into the text in the manuscript on Page 17 – Line 372 -374.
Comment 4: Is there any problem for Figure 1?
Response 4: Looks like figure 1 was missing from the PDF file even though this was present in our submitted version. We made sure all the figures appear in PDF version of the manuscript. Page 4 -Line 138.
Comment 5: Please explain “ppm” and “ppb” when first use.
Response 5: The full forms of these abbreviations/units have been added at their first place of occurrence as well as in footnotes of Table 1. ppm is parts per million or mg/kg and ppb is parts per billion or µg/kg – revised on Page 3-Line 122 and Page 9, footnotes of Table 1 .
Comment 6: Line 447: What kind of rice did you used for preparing rice media? Did you buy it from supermarket? Did you used the same rice throughout the research? Do you the protein and starch contents of the rice used?
Response 6: Thanks for the comments. We provided the source and type of rice in the Page 19, Line 468-469. It was Organic long grain white rice, and we used the same rice throughout the research. We did not evaluate the protein and starch contents of the rice.
Comment 7: Please provide the company name of PDA used.
Response 7: : Thanks for the comment. We provided the PDA information in the page 17, Line 378.
Reviewer 3 Report
Comments and Suggestions for Authors
The manuscript deals with the identification of 3ANX and NX producing isolates of F. graminearum in Western Canada, and it is the first marking such a large collection of isolates.
It is certainly a novel study with a large impact on the current knowledge in plant pathology. Such a study can provide insights into the F. graminearum chemotyping also with respect to climate change. It is therefore, in my view, worth of publication.
The adopted methodology is very sound and properly presented, the results are convincing, and conclusions are supported by evidence.
As a weakness, the huge amount of data provided could be difficult to catch at the first read, and their implication for the field could be not immediately clear to a non-expert reader. Relevance for the scientific community is only discussed shortly at the end of the discussion section, while it should be more widely addressed throughout the manuscript.
In addition, I would like to see discussed the potential correlation among 3ANX/NX chemotypes prevalence and hosting crops, if any? Is there any pattern in the isolation from the field? And is the NX/3ANX production somehow modulated by the hosting crop compared to what observed in vitro? Any evidence to be discussed?
Similarly, did the authors observed any correlation between the chemotype prevalence and the climate over time? Is this something that could be addressed at a later stage?
As a minor suggestion, I would therefore invite the authors to further address such topics and anyway to deepen the discussion, in order to make clearer the impact of their findings to the field.
Author Response
Comment 1: The manuscript deals with the identification of 3ANX and NX producing isolates of F. graminearum in Western Canada, and it is the first marking such a large collection of isolates.
It is certainly a novel study with a large impact on the current knowledge in plant pathology. Such a study can provide insights into the F. graminearum chemotyping also with respect to climate change. It is therefore, in my view, worth of publication.
Response 1: Thank you and we greatly appreciate for recognizing the large collection of isolated that were tested in the manuscript. We also appreciate for recognizing the importance of this research work in plant pathology. Thank you for providing valuable comments and suggestions which helped improve the manuscript. All the comments have been addressed as mentioned below.
Comment 2: The adopted methodology is very sound and properly presented, the results are convincing, and conclusions are supported by evidence.
Response 2: Thank you.
Comment 3: As a weakness, the huge amount of data provided could be difficult to catch at the first read, and their implication for the field could be not immediately clear to a non-expert reader. Relevance for the scientific community is only discussed shortly at the end of the discussion section, while it should be more widely addressed throughout the manuscript.
Response 3: Thanks for your comment and we agree. We added additional text in the abstract and introductions section to address this comment. In the abstract: [Page 1, line 27-31]. In the introduction [Page 3, line 98-102].
Comment 4: In addition, I would like to see discussed the potential correlation among 3ANX/NX chemotypes prevalence and hosting crops, if any? Is there any pattern in the isolation from the field? And is the NX/3ANX production somehow modulated by the hosting crop compared to what observed in vitro? Any evidence to be discussed?
Response 4: Thanks for the comments. The next phase of the study is on-going and the potential correlation among 3ANX/NX chemotypes prevalence and hosting crops in-plant experiments (ie; wheat, barley, corn, oats) are being investigated. Current research was performed in-vitro from F. graminearum isolates collected only from wheat, so the data doesn’t support any potential correlation among 3ANX/NX chemotypes prevalence or NX/3ANX modulated by the hosting crop. It has been reported ( included in the current manuscript) that 3ANX toxin is typically deacetylated into NX in plants; however, such deacetylation was incomplete in our in-vitro cultures and as a result, NX concentrations ranging from 0.07 to 3.2 ppm were observed. While it is outside the scope of current manuscript, we hope to generate data and discuss isolation patterns and host crop influences on NX/3ANX in our future publications.
Comment 5: Similarly, did the authors observed any correlation between the chemotype prevalence and the climate over time? Is this something that could be addressed at a later stage?
Response 5: The point highlights a very important aspect and we thank you for that. While it is out of scope for this manuscript, we would like to share that this study is on-going and impact of climate change are being investigated. Results of those studies will be published upon completion of studies.
Comment 6: As a minor suggestion, I would therefore invite the authors to further address such topics and anyway to deepen the discussion, in order to make clearer the impact of their findings to the field.
Response 6: We added additional text on the potential biosynthesis pathways for these NX toxins as well as synthesis of more than one mycotoxin by a chemotype. in the abstract and introductions section to address this comment. In the abstract: [Page 16, lines 336-347. In addition, we also added additional text in the abstract [Page 1, line 27-31] and the introduction [Page 3, line 98-102] to highlight the impact of our findings.